# The Development of Musculoskeletal Disorders during Undergraduate Dentistry Studies—A Long-Term Prospective Study

**DOI:** 10.3390/ijerph18147662

**Published:** 2021-07-19

**Authors:** Martin Kapitán, Lenka Hodačová, Eva Čermáková, Stanislav Machač, Jan Schmidt, Nela Pilbauerová

**Affiliations:** 1Department of Dentistry, Charles University, Faculty of Medicine in Hradec Králové, and University Hospital Hradec Králové, 500 05 Hradec Králové, Czech Republic; jan.schmidt@lfhk.cuni.cz (J.S.); Nela.Pilbauerova@lfhk.cuni.cz (N.P.); 2Department of Preventive Medicine, Charles University, Faculty of Medicine in Hradec Králové, 500 03 Hradec Králové, Czech Republic; HodacovaL@lfhk.cuni.cz; 3Department of Medical Biophysics, Charles University, Faculty of Medicine in Hradec Králové, 500 03 Hradec Králové, Czech Republic; cermakovae@lfhk.cuni.cz; 4Department of Rehabilitation and Sports Medicine, Charles University, Second Faculty of Medicine, and University Hospital Motol, 150 06 Prague, Czech Republic; machac.s@seznam.cz; 5Institute of Sports Medicine, 150 00 Prague, Czech Republic

**Keywords:** dental ergonomics, dentistry students, development, longitudinal prospective evaluation, musculoskeletal disorders, questionnaire, risk factors

## Abstract

Musculoskeletal disorders (MSDs) frequently occur among dental practitioners and present a significant occupational burden with an early onset in the dentists’ career. This study aimed to analyze the five-year development of self-reported overall MSDs among the dentistry students during the course of their studies and to assess the possible influence of the risk as well as protective factors. The questionnaire inquiry was performed among the first-year dentistry students, regarding the occurrence of MSDs and the presence of potential risk and protective factors. The same students were followed, and they filled in the same questionnaire in the middle and at the end of their studies. A total of 73 dentistry students and 28 general medicine students participated. The occurrence of the overall MSDs statistically significantly increased from 30.1% at the beginning of the first year to 45.2% at the end of the fifth year among the dentistry students. The top-level sport was statistically significantly associated with the higher occurrence of MSDs in the fifth year and with the development of new MSDs between the first and the fifth year. This longitudinal prospective evaluation showed a significant increase in the MSDs occurrence among the dentistry students during their studies.

## 1. Introduction

Musculoskeletal disorders (MSDs) are defined as a group of diseases with various symptoms affecting different parts of a locomotive system. They manifest mainly with pain, muscle stiffness, restricted range of motions, and decreased muscular strength [1].

There is broad evidence of an association between work-related MSDs [2] and the dentistry profession [3,4,5,6,7]. The occurrence of MSDs among dentists varies between 47% and 93% in different studies [3,8,9,10,11,12,13,14,15,16,17,18,19,20,21,22,23,24]. Several studies among dentistry students showed the early onset of MSDs even during the undergraduate studies, where the occurrence ranged from 39% to 93% [8,25,26,27,28,29,30,31,32]. The comparative studies showed a higher MSDs occurrence in later years of studies than in earlier phases [27,33]. However, these studies compared actual students in different years of dentistry studies. In all the studies, women suffered from MSDs more often than men [3,8,9,14,15,16,20,23,25,26,27,28,29,34,35,36]. The most frequent areas of pain are neck, lower back, shoulders and hands [3,7,14,15,16,17,19,20,23,24,26,27,28,29,32,35,36,37,38,39,40,41,42]. The occupational risk factors in dentistry include long-term static asymmetric forced posture, small intensively lighted working field, limited range and spectrum of movements, noise, and psychosocial stress [12,23,35,38,41,43,44,45,46,47,48,49,50,51]. Although the ergonomic recommendations have been developed and their efficacy was proved [52], they seem not to be broadly known [37], or they are ignored in everyday practice [7,45].

The dentistry study program lasts for five years at the Charles University, Faculty of Medicine in Hradec Králové, the Czech Republic. The first two years consist of preclinical subjects with practical lessons on various simulators and models (a total of 300 teaching hours). The students start their clinical practice in the winter semester of the third year when they acquaint basic workflow of the dental office and train elementary procedures among each other, such as local anesthesia, the rubber dam placement, and taking the intraoral X-rays (a total of 60 teaching hours). During the second half of the studies, i.e., five semesters, the students have clinical practical lessons twice a week at different departments of the dentistry clinic (a total of 975 teaching hours). Concurrently, the dentistry students go through the majority of theoretical subjects and general medicine subjects, similarly to their general medicine colleagues. The total number of teaching hours is higher in the dentistry students.

The education in dental ergonomics consists of one theoretical lecture of three teaching hours in the first year and one dedicated practical lesson with the simulators. During all preclinical and clinical practical lessons, the students are explained and instructed in ergonomic recommendations and encouraged to perform compensation exercises. The dental chairs used during the clinical practical lessons can be rearranged for the left-handed dentist.

As far as the authors are aware, there are no studies in the available literature that would observe the development of self-reported MSDs among one group of dentistry students during the whole length of their studies.

This study aimed to analyze the five-year development of self-reported overall MSDs among the dentistry students during the whole studies and to assess the possible influence of the risk as well as protective factors on the development of musculoskeletal pain in the early stages of the dentistry career.

## 2. Materials and Methods

The dentistry students at the Charles University, Faculty of Medicine in Hradec Králové, the Czech Republic, were observed during the whole length of their study, i.e., for five years. All the students admitted to the dentistry studies in the academic years 2014/15, 2015/16, and 2016/17 were invited to participate in this study. The inclusion criterion was the enrollment to the dentistry studies in one of the aforementioned academic years. The exclusion criterion was no previous university study in the same study program. No student met this criterion. Students of general medicine were offered to participate as a control group, with the same inclusion (enrollment to general medicine studies) and exclusion criteria. Another exclusion criterion applied in the further phases of the study was any irregularity in the course of the studies, i.e., individual study plan, repeated year, an internship abroad, etc. Thus, only the students with the regular running and length of the studies were followed.

In the initial phase of the longitudinal prospective evaluation, the participants filled in the questionnaire and were objectively examined using the Spinal Mouse^®^ device (Idiag AG, Fehraltorf, Switzerland). The participation was voluntary; all the respondents signed the informed consent.

The same students were followed during their studies and were asked to fill in the same questionnaire again in the middle of the third study year and at the end of the fifth year. At the same time, they were objectively examined using the Spinal Mouse^®^ device (Idiag AG, Fehraltorf, Switzerland) again.

The questionnaire was developed by the authors based on the Nordic standardized questionnaire [53] and questionnaires used by the authors’ team in a similar study among dentists [23,35]. The same questionnaire was used in our short-term study [27], where a part of the respondents overlapped with the current sample.

The questionnaire consisted of three parts. The first part concerned the personal data, such as gender, age, height, and body weight. The second group of questions followed regarding the MSDs, possible risk and protective factors, ways of dealing with pain, and other opinions related to MSDs. The last part included questions about the intensity of pain in different body regions according to the Nordic standardized questionnaire [53].

The method was approved by the Ethics Committee of the University Hospital Hradec Králové (Ref. no. 201410 S04P) and by the Dean of the Charles University, Faculty of Medicine in Hradec Králové.

The quantitative data are presented by means and standard deviations, the qualitative data by counts and percentage counts. The differences between the groups were analyzed by Kolmogorov–Smirnov test for the quantitative data and by the chi-square test or Fisher’s exact test for the qualitative parameters. The development between the years of study was analyzed by Wilcoxon paired test for the quantitative data and by McNemar test for the qualitative parameters. Univariate and multivariate logistic regression was used to evaluate the influence of the followed factors on the occurrence and development of self-reported overall MSDs. Cohen’s g and common language effect size (CLES) were calculated and presented as measures of the size of the effect for the results, which were statistically significant and relevant as the defined goals of the study.

The level of statistical significance was set to α = 0.05. NCSS 2019 Statistical Software (NCSS, LLC, Kaysville, UT, USA) was used.

## 3. Results

A total of 100 dentistry (D) students and 73 general medicine (GM) students entered the first phase of the study. Some of them did not volunteer to continue in the study or were excluded for meeting the exclusion criteria. Finally, 73 dentistry students (73%) and 28 general medicine students (38%) remained.

The gender distribution, age, and body parameters of the participants are presented in Table 1. The values referring to the significance of differences between the three phases of the study and between the study groups are listed.

Table 2 summarizes the presence of the potential risk and protective factors for MSDs among the respondents. There were no statistically significant differences between the study groups in any factor in any study year, as well as no statistically significant differences between the three phases of the study.

Subjective assessment of the general health status of the respondents is summarized in Figure 1.

The development of the MSDs occurrence among the whole groups of respondents is demonstrated in Figure 2. The increase from the first year to the fifth year among the D students was statistically significant (*p* = 0.041; Cohen’s g = 0.19; CLES = 62.21%). Among the GM students, the increase was not statistically significant (*p* = 0.15). The differences of MSDs among genders in the dentistry students are described in Table 3. No statistical significances were proven in any of the followed years of studies.

The occurrence of recent pain in different body regions is presented in Table 4. The answers “mild”, “moderate”, and “severe” were joined.

Table 5 shows the ways of solving the pain by the respondents and their perception of the influence of MSDs on their lifestyle and the studies’ influence on MSDs.

A total of 47.9% of the dentistry students (*n* = 35) declared, at the beginning of their studies, they had known that two-thirds of dentists suffer from MSDs. In contrast, only 3.6% of the general medicine students (*n* = 1) had had this information. This difference is statistically significant (*p* < 0.001).

The range of the ergonomic education was considered as sufficient by 80.8% of the dentistry students in the third year (*n* = 59) and 78.1% (*n* = 57) in the fifth year, respectively.

The univariate analysis did not find in our sample a statistically significant influence of any factor on the MSDs occurrence or development (increase or decrease), except for the age in the first year, the top-level sport in the fifth year, and the regular sporting activity in the third year. The age of the students statistically significantly influenced the occurrence of MSDs in the first year, but not in the following years of studies. The top-level sport was statistically significantly associated with the higher occurrence of MSDs in the fifth-year students and with the development of new MSDs from the first year to the fifth year. The details about the effects of the followed factors on the MSDs occurrence are shown in Table 6; Table 7 presents the influence of the followed factors on the changes of MSDs (newly developed or disappeared) from the first year to the fifth year of studies.

The results of the multivariate analysis confirmed the statistically significant influence of the age and top-level sport, but not the regular sporting activity, on the MSDs occurrence. Additionally, the multivariate analysis revealed the statistically significant influence of the presence of diseases of the musculoskeletal system in blood relatives on the MSDs occurrence in students in the first year. There was no statistically significant influence of any of the followed factors on the newly developed MSDs found in the multivariate analysis. The detailed results of the multivariate analysis are presented in Table 8 and Table 9.

The analysis of the main problematic body regions concerning MSDs in students and the findings from the physical examination will be published in the following papers.

## 4. Discussion

This study aimed to investigate the development of MSDs among the dentistry students during the undergraduate studies. The general medicine students were intended to be employed as a control group. A total of 100 dentistry students and 73 general medicine students entered the study. A significant number of the participants did not finish the follow-up because they decided not to continue or were excluded. This decrease happened mainly among the group of general medicine students, where only 28 of them remained. Such a limited group was not sufficient for a planned analysis of the influence of the followed factors on MSDs and couldn‘t function as a control group as was planned. The comparison of both groups was thus made only in terms of the participants characteristics and the presence of the followed factors. Other results of the general medicine students are even though presented to illustrate the overall situation.

The gender distribution was the only statistically significant difference between the dentistry students and the general medicine students. There were many more women among the dentistry students, whereas the general medicine group was more balanced in our sample. This corresponds with the traditional gender distribution among dentists in the Czech Republic. According to the last yearbook of the Czech Dental Chamber, there were 35.1% of male and 64.9% of female dentists [54]. The difference in our dentistry students is even larger; however, the reasons are not known. There were no other differences found between the study group and the control group in personal characteristics nor in the presence of possible risk and protective factors.

There was a statistically significant increase in the occurrence of self-reported MSDs among the dentistry students, from 30.1% at the beginning of the first year to 45.2% at the end of the fifth year. The Cohen’s g of 0.19 indicates a medium size of the effect [55]. The increase among the general medicine students was not statistically significant, even if it was relatively similar, around 50% from the baseline. The small size of the control group probably contributed; thus, the effective comparison of the MSDs development between the D students and GM students is not possible.

We believe that there are two main reasons for the higher rate of MSDs in the general medicine students and for the increase between the first and the fifth year: (i) the general medicine students were less aware of this problem before they came to the university compared with the dentistry students. We assume that people considering dentistry to be their further profession were able to find relevant information and to decide responsibly, i.e., if they already suffer from any musculoskeletal problem, they may more likely choose a different kind of expertise. This might have contributed to a higher rate of occurrence of MSDs among GM students in our study, even in the first year when it could be expected that the students of both groups did not differ in their previous study experience and presence of related risk factors; (ii) as the dentistry students were aware of the MSDs occurrence among dentists, they had a chance to apply preventive matters, which, in combination with a declared sufficient range of ergonomic education, seem to be at least partly successful. Thus, we suggest a conclusion that the initial awareness of the problem, sufficient ergonomic education, and the application of ergonomic recommendations compensated the risk factors involved during the dentistry studies, which led to similar dynamics of the MSDs occurrence increase as in the general medicine students.

The answers to questions about recent pain in different body regions were joined and are presented regardless of the pain intensity. The reason is that quite a small number of the answers were “moderate” and “severe.” In a future study, these data will be analyzed in detail and prepared for publication.

In accordance with other studies among dentistry students and dentists, there were more MSDs in women than in men; however, this finding was not statistically significant in our study. The overall level of MSDs occurrence in the dentistry students was relatively small in comparison with other studies in the literature [8,25,26,27,28,29,30,31,32]. There seem to be several reasons for this positive finding.

The students have spent just a short time on the dental work. During the studies, they work only part time in the phantom labs and the dental offices. Till the end of the studies, they have had only 2.5 years of clinical work behind them. Thus, the risk factors contributing to MSDs in the dentistry profession did not act for a sufficient time.

Another factor was the young age of the students, between 19 and 24 years. Thus, we can presume a sufficient regenerative potential, body flexibility, and adaptability to physical loading. Additionally, most of the factors causing the MSDs in dentists were not yet present, i.e., they could not have played a role yet in the students. According to our study among dentists in the Czech Republic [35], these factors causing musculoskeletal pain were the age, the length of practice, more than 20 patients a day, considering the work as psychically demanding, and the assessment of the general health status as bad/very bad. The majority of the students of both groups considered the studies as psychically demanding. This question was used to identify stress arising from the studies subjectively perceived by the students. The other factors will act later in the dentistry career.

Most students considered the current level of ergonomics education sufficient. They were able to apply the ergonomics recommendations both in preclinical and clinical practice. It might have contributed to a rather smaller occurrence of MSDs among the respondents. In some students, the originally present MSDs even disappeared during the running of the studies. It could have been caused by the initial awareness of this problem in the dentistry occupation resulting in paying more attention to ergonomics during the studies.

The increase in the number of dentistry students solving the pain by exercising between the first and the fifth year was found to be the only statistically significant factor among the ways how students solved MSDs. It is positive that the students almost did not use analgesics; it indicates just a mild intensity of musculoskeletal pain. Another positive finding is that many of the students tend to seek help of a specialist. Four respondents out of the whole sample (101 students) underwent surgery for MSDs, which is quite a lot with respect to young age. Among the Czech dentists in our previous study, 86.9% of them solved the MSDs by exercising, 48.6% used self-prescribed analgesics, 46.7% sought help of a specialist, and 5.8% underwent surgery [23].

The perception of the influence of MSDs on the lifestyle slightly increased in both groups; however, it was not statistically significant. The same trend was observed between the third year and the fifth year in terms of considering the influence of the studies on the occurrence of MSDs.

Almost all students in both groups declared very good or good general health status, only two of them declared satisfactory health status, and no one reported bad or very bad health status. This shows that young people who choose dentistry as their profession are healthy. The reason may be associated with the awareness of the MSDs among dentists discussed above. Good general health, even in the fifth year, could have contributed to the relatively small occurrence of MSDs. A significant influence of self-reported general health status was proved in our study among the Czech dentists, where the perceived satisfactory or bad/very bad general health was the strongest risk factor for MSDs [35].

Out of the followed risk and protective factors, the age and the top-level sport had a statistically significant influence on MSDs according to both univariate and multivariate analysis. The age influenced the occurrence of MSDs in the first year only. As the age of the students was similar, the relevance of this finding is unclear.

As a top-level sport, we understand a sporting activity that is organized in sport clubs, includes regular workouts, practice, matches, and competitions, is performed on a long-term basis, and the athletes are registered by the official, state-controlled body. The top-level sport increased the occurrence of MSDs in the fifth year more than five times, and the risk of the development of new MSDs between the first year and the fifth year more than four times. Thus, it can be recommended not to study dentistry for people doing a top-level sport currently or in the past.

The presence of diseases of the musculoskeletal system in blood relatives influenced the occurrence of MSDs only in the first-year students, whereas later during the dentistry studies, its influence was not statistically significant. This finding is in accordance with our previous study among dentists, which did not prove a statistically significant relationship between this factor and the occurrence of musculoskeletal pain [23,35]. Thus, we assume that diseases of the musculoskeletal system in blood relatives are not a factor that should prevent young people from dentistry studies.

The analysis of the influence of regular sporting activity on the MSDs brought confusing results. The regular sport was found a risk factor of MSDs in the first and the fifth year but without a statistical significance. In the third year, on the contrary, there was a statistically significant decrease in the occurrence of MSDs. This influence was not statistically significant in the multivariate analysis. These contradictory results may have been caused by several reasons, such as inconsistent answers of the respondents, uneven distribution of performing the sporting activities during the seasons of the year, as well as variable volume of demanded study activities in different periods of the academic year. The three phases of the study were carried out in October (the first year), February (the third year), and May (the fifth year). In the scientific literature, there is not a general agreement on the benefits of sporting activities in relation to MSDs among dentists nor dentistry students [12,14,22,23,24,25,27,33,35].

The multivariate analysis was not calculated in the group of students, where the original MSDs disappeared because of its limited size.

The other followed risk factors, which were proved to have an influence on MSDs in dentists, did not show such influence in the case of dentistry students in our study. The reason probably is that some of the factors (work in a forced, unnatural position, small working field, long-term focus on a short distance, etc.) have not acted for a sufficient time yet, as the students have passed only a negligible volume of the clinical practice, compared to the dentists. Some other factors were not yet even present in students, e.g., general health conditions, long working hours, stress from running a private practice, the experience of disease or trauma of the musculoskeletal system, etc.

The noteworthy positive finding is that no respondent smoked during the studies. There were only two smokers in the whole sample of 101 students in the first year, who stopped till the third year. The studies indicate between 8.5% and 17% of smokers among physicians in the Czech Republic [56,57]. Another notable fact is the significant increase in the participants’ weight during the studies. It probably resulted from difficult studies both in dentistry and general medicine with an enormous volume of theoretical knowledge, which the students must learn, and correspondingly lack of time available for physical activities.

The limits of our study arise from the relatively small number of respondents, particularly in the control group. Our faculty is rather small, with the regular number of students admitted to the dentistry study program around 35 every year. Thus, the authors endeavored to increase the sample size by involving students of three subsequent academic years. Almost all of them entered the study; however, some had to be excluded for not regular running of the studies. In the control group, many students left the general medicine studies; others denied continuing in the follow-up. The planned comparison of the MSDs occurrence and development was not possible. Nevertheless, the results show the development of MSDs during dentistry studies and can roughly illustrate the trends among the general medicine students. Even though the control group was not representative, we find it important to follow both groups, as they are similar in terms of other characteristics.

Another limitation is that the data are based on the subjective statements of individual respondents. However, this way of acquiring information is common in similar studies, as being a cheap, fast, and sufficiently reliable method. The consistency of the results is increased by asking the same respondents three times and not comparing the answers of different people, which was the study design of our previous short-time study [27]. The same questionnaire was used.

The advantage of our study is the long-term follow-up of the students during the whole length of their undergraduate studies. As the authors are aware, other up-to-date available studies with similar topic analyzed only the current status at one particular time or compared groups of different students. Another advantage is the investigation of influence of several possible factors on the development of MSDs in the early stage of the dentistry career. We believe the gained data can contribute to better understanding of this development. Additionally, our results can provide a baseline for further research in this field.

Moreover, the results can be effectively compared with our previous study among the Czech dentists [23,35] because the work and living environment are the same, and the study conditions are similar, at least for the younger group of dentists.

Further research of the development of MSDs in the early stages of the dentistry career is necessary. The subsequent follow-up of our sample is planned in the future.

## 5. Conclusions

Almost half of the dentistry students suffered from MSDs just before graduation.

The occurrence of MSDs among the dentistry students increased significantly from the 1st year to the 5th year of study.

The top-level sport significantly contributed to the occurrence and the development of MSDs. No statistically significant influence of all the other followed factors (gender, height, actual body weight, changes in body weight, general disease, chronic medication, smoking, dominant hand, diseases of the musculoskeletal system in blood relatives, past disease or trauma of musculoskeletal system, regular sporting activities, and considering the studies as psychically demanding) was demonstrated in our sample of the dentistry students.

## Figures and Tables

**Figure 1 ijerph-18-07662-f001:**
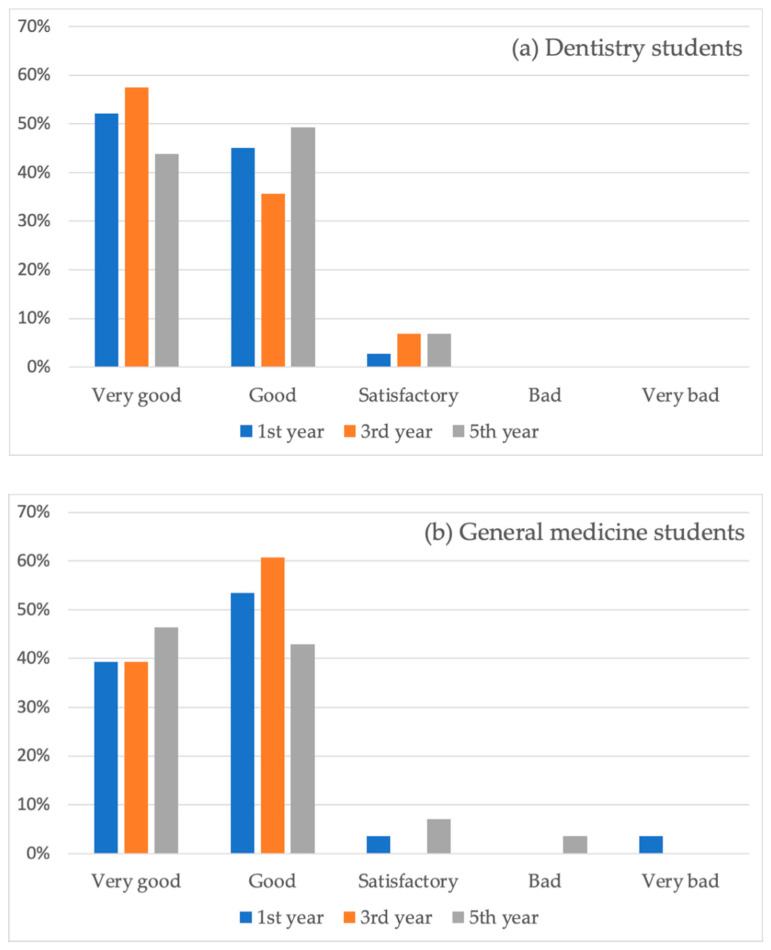
The declared general health status of the dentistry students (**a**) and general medicine students (**b**).

**Figure 2 ijerph-18-07662-f002:**
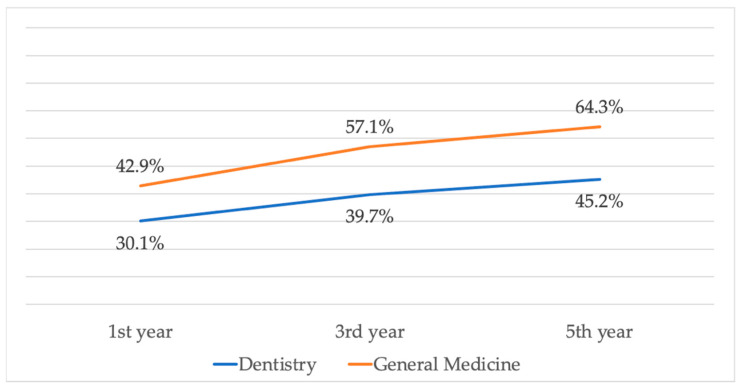
The development of MSDs during the running of the studies.

**Table 1 ijerph-18-07662-t001:** Sample characteristics.

		Dentistry	General Medicine	D vs. GM *
		% (*n*)	% (*n*)	
Gender	Men	27.4 (20)	53.6 (15)	*p* = 0.019
	Women	72.6 (53)	46.4 (13)	
		mean (SD)	mean (SD)	
Age	1st year	19.6 (1.11)	19.2 (0.42)	*p* = 0.28
Height -men	1st year	181.2 (8.38)	181.8 (7.15)	*p* = 0.82
Weight -men	1st year	75.9 (13.45)	78.9 (12.67)	*p* = 0.53
		1st vs. 3rd: *p* = 0.0025	1st vs. 3rd: *p* = 0.015	
	3rd year	78.8 (13.81)	82.1 (15.7)	*p* = 0.53
		3rd vs. 5th: *p* = 0.88	3rd vs. 5th: *p* = 0.00094	
	5th year	78.9 (13.7)	86.1 (16.7)	*p* = 0.18
		1st vs. 5th: *p* = 0.014	1st vs. 5th: *p* = 0.00077	
BMI -men	1st year	23.0 (2.95)	23.9 (3.91)	*p* = 0.72
		1st vs. 3rd: *p* = 0.0019	1st vs. 3rd: *p* = 0.012	
	3rd year	24.0 (2.72)	24.8 (4.90)	*p* = 0.73
		3rd vs. 5th: *p* = 0.89	3rd vs. 5th: *p* = 0.0016	
	5th year	24.0 (2.90)	26.0 (5.09)	*p* = 0.47
		1st vs. 5th: *p* = 0.011	1st vs. 5th: *p* = 0.0083	
Height -women	1st year	167.9 (5.83)	167.6 (6.12)	*p* = 0.86
Weight -women	1st year	59.1 (6.42)	59.2 (6.83)	*p* = 0.99
		1st vs. 3rd: *p* = 0.47	1st vs. 3rd: *p* = 0.040	
	3rd year	59.5 (6.97)	61.2 (7.61)	*p* = 0.45
		3rd vs. 5th: *p* = 0.00027	3rd vs. 5th: *p* = 0.13	
	5th year	60.8 (7.34)	62.0 (8.01)	*p* = 0.41
		1st vs. 5th: *p* = 0.00046	1st vs. 5th: *p* = 0.018	
BMI -women	1st year	20.7 (2.55)	21.1 (2.35)	*p* = 0.75
		1st vs. 3rd: *p* = 0.35	1st vs. 3rd: *p* = 0.073	
	3rd year	21.0 (1.99)	21.7 (2.35)	*p* = 0.41
		3rd vs. 5th: *p* = 0.0007	3rd vs. 5th: *p* = 0.44	
	5th year	21.5 (2.21)	22.1 (2.60)	*p* = 0.56
		1st vs. 5th: *p* = 0.00071	1st vs. 5th: *p* = 0.022	

* The column “D vs. GM” shows the *p* values for the comparison between the dentistry students and general medicine students.

**Table 2 ijerph-18-07662-t002:** The presence of monitored risk and protective factors among the respondents.

	1st Year		3rd Year		5th Year	
	D	GM	D	GM	D	GM
	% (*n*)	% (*n*)	% (*n*)	% (*n*)	% (*n*)	% (*n*)
General disease	15.1 (11)	10.7 (3)	15.1 (11)	7.1 (2)	21.9 (16)	25 (7)
Chronic medication	24.7 (18)	25 (7)	23.3 (17)	25 (7)	16.4 (12)	32.1 (9)
Smoking	1.4 (1)	3.6 (1)	0 (0)	0 (0)	0 (0)	0 (0)
Dominant hand-right	90.4 (66)	92.9 (26)	91.8 (67)	96.4 (27)	91.8 (67)	96.4 (27)
Diseases of musculoskeletal system in blood relatives	45.2 (33)	53.6 (15)	35.6 (26)	39.3 (11)	39.7 (29)	50 (14)
Past disease or trauma of musculoskeletal system	9.6 (7)	7.1 (2)	19.2 (14)	7.1 (2)	16.4 (12)	21.4 (6)
Top-level sport	21.9 (16)	14.3 (4)	16.4 (12)	25 (7)	17.8 (13)	17.9 (5)
Regular sporting activities	78.1 (57)	85.7 (24)	86.3 (63)	96.4 (27)	79.5 (58)	82.1 (23)
Considering the studies as psychically demanding			97.3 (71)	96.4 (27)	97.3 (71)	96.4 (27)

**Table 3 ijerph-18-07662-t003:** The occurrence of MSDs among male and female dentistry students.

	Male	Female	
	% (*n*)	% (*n*)	*p*
1st year	22.2 (4)	32.7 (18)	0.56
3rd year	38.9 (7)	40.0 (22)	1
5th year	33.3 (6)	49.1 (27)	0.29

**Table 4 ijerph-18-07662-t004:** The occurrence of recent pain in different body regions regardless of its intensity.

	1st Year		3rd Year		5th Year	
	D	GM	D	GM	D	GM
	% (*n*)	% (*n*)	% (*n*)	% (*n*)	% (*n*)	% (*n*)
Head	42.5 (31)	42.9 (12)	34.2 (25)	42.9 (12)	47.9 (35)	35.7 (10)
Neck	46.6 (34)	50.0 (14)	52.1 (38)	46.4 (13)	61.6 (45)	53.6 (15)
Upper back	23.3 (17)	35.7 (10)	32.9 (24)	46.4 (13)	32.9 (24)	28.6 (8)
Lower back	42.5 (31)	50 (14)	47.9 (12)	57.1 (16)	49.3 (36)	67.9 (19)
Shoulders	16.4 (12)	32.1 (9)	23.3 (17)	17.9 (5)	24.7 (18)	28.6 (8)
Elbows	2.7 (2)	3.6 (1)	4.1 (3)	7.1 (2)	5.5 (4)	3.6 (1)
Wrists/hands	8.2 (6)	10.7 (3)	20.5 (15)	7.1 (2)	26.0 (19)	17.9 (5)
Hips/tights	13.7 (10)	21.4 (6)	16.4 (12)	7.1 (2)	16.4 (12)	17.9 (5)
Knees	21.9 (16)	28.6 (8)	27.4 (20)	25 (7)	24.7 (18)	28.6 (8)
Ankles/feet	5.5 (4)	25.0 (7)	4.1 (3)	14.3 (4)	6.8 (5)	7.1 (2)

**Table 5 ijerph-18-07662-t005:** The ways of solving of MSDs and associated influences.

	1st Year		3rd Year		5th Year	
	D	GM	D	GM	D	GM
	% (*n*)	% (*n*)	% (*n*)	% (*n*)	% (*n*)	% (*n*)
Exercising	23.3 (17) *	35.7 (10)	30.1 (22)	39.3 (11)	41.1 (30) *	46.4 (13)
Analgesics	1.4 (1)	0 (0)	0 (0)	0 (0)	1.4 (1)	3.6 (1)
Intervention by a specialist	9.6 (7)	17.9 (5)	6.9 (5)	7.1 (2)	8.2 (6)	3.6 (1)
Surgery	0 (0)	3.6 (1)	0 (0)	0 (0)	1.4 (1)	7.1 (2)
Perceived influence of MSDs on the lifestyle	5.5 (4)	7.1 (2)	8.2 (6)	10.7 (3)	9.6 (7)	14.3 (4)
Perceived influence of studies on MSDs			15.1 (11)	35.7 (10)	26 (19)	39.3 (11)

* *p* = 0.019 (for the dentistry students between the first year and the fifth year).

**Table 6 ijerph-18-07662-t006:** The univariate analysis of the influence of the followed factors on the occurrence of MSDs among the dentistry students.

	1st Year		3rd Year		5th Year	
	OR (95% CI)		OR (95% CI)		OR (95% CI)	
Gender (women)	1.70 (0.49, 5.92)	*p* = 0.39	1.05 (0.35, 3.12)	*p* = 0.93	1.93 (0.63, 5.87)	*p* = 0.24
Age	1.94 (1.05, 3.62)	*p* = 0.011	1.09 (0.74, 1.59)	*p* = 0.67	0.94 (0.64, 1.39)	*p* = 0.75
	CLES = 60.19%					
Height-men	1.15 (0.96, 1.39)	*p* = 0.084	0.97 (0.86, 1.09)	*p* = 0.57	1.04 (0.92, 1.18)	*p* = 0.49
Height-women	0.95 (0.86, 1.05)	*p* = 0.29	1.02 (0.93, 1.12)	*p* = 0.63	1.04 (0.95, 1.14)	*p* = 0.36
Actual weight-men	1.05 (0.96, 1.15)	*p* = 0.26	0.97 (0.90, 1.05)	*p* = 0.41	1.00 (0.93, 1.07)	*p* = 0.93
Actual weight-women	1.00 (0.92, 1.10)	*p* = 0.94	0.96 (0.88, 1.04)	*p* = 0.28	1.02 (0.95, 1.10)	*p* = 0.57
Weight increase			0.60 (0.20, 1.81)	*p* = 0.65	1.30 (0.48, 3.55)	*p* = 0.70
Weight decrease			0.92 (0.25, 3.43)		1.88 (0.37, 9.52)	
General disease	0.45 (0.12, 1.68)	*p* = 0.24	1.18 (0.31, 4.47)	*p* = 0.80	1.08 (0.35, 3.29)	*p* = 0.90
Chronic medication	2.34 (0.77, 7.11)	*p* = 0.14	0.56 (0.17, 1.79)	*p* = 0.31	0.84 (0.24, 2.95)	*p* = 0.79
Dominant hand (right)	1.09 (0.19, 6.08)	*p* = 0.92	0.63 (0.12, 3.38)	*p* = 0.60	0.81 (0.15, 4.31)	*p* = 0.81
Diseases of musculoskeletal system in blood relatives	2.24 (0.81, 6.20)	*p* = 0.12	1.51 (0.57, 4.01)	*p* = 0.41	1.55 (0.60, 3.98)	*p* = 0.36
Past disease or trauma of musculoskeletal system	0.92 (0.16, 5.15)	*p* = 0.92	1.17 (0.36, 3.82)	*p* = 0.79	1.26 (0.36, 4.35)	*p* = 0.72
Top-level sport	1.54 (0.48, 4.93)	*p* = 0.47	1.65 (0.48, 5.73)	*p* = 0.43	5.36 (1.33, 21.55)	*p* = 0.010
					CLES = 74.36%	
Regular sporting activities	2.17 (0.55, 8.54)	*p* = 0.25	0.23 (0.05, 0.98)	*p* = 0.037	1.31 (0.41, 4.15)	*p* = 0.65
			CLES = 71.67%			
Awareness of MSDs among dentists	1.90 (0.69, 5.25)	*p* = 0.21	0.64 (0.25, 1.66)	*p* = 0.36	0.83 (0.33, 2.10)	*p* = 0.70
Considering the range of ergonomic education as sufficient			0.59 (0.18, 1.92)	*p* = 0.39	0.57 (0.18, 1.73)	*p* = 0.32

CLES = common language effect size.

**Table 7 ijerph-18-07662-t007:** The univariate analysis of the influence of the followed factors on the newly developed and disappeared MSDs among the dentistry students.

	From the 1st Year to the 5th Year			
	Newly Developed MSDs		Disappeared MSDs	
	OR (95% CI)		OR (95% CI)	
Gender (women)	1.90 (0.50, 7.20)	*p* = 0.33	0.64 (0.07, 5.61)	*p* = 0.69
Age	0.82 (0.42, 1.59)	*p* = 0.55	1.10 (0.67, 1.81)	*p* = 0.70
Height-men	1.02 (0.88, 1.17)	*p* = 0.83	0.67 (0.19, 2.32)	*p* = 0.20
Height-women	1.06 (0.94, 1.19)	*p* = 0.35	0.95 (0.80, 1.11)	*p* = 0.49
Actual weight-men	0.99 (0.91, 1.08)	*p* = 0.90	1.02 (0.84, 1.24)	*p* = 0.84
Actual weight-women	1.04 (0.94, 1.15)	*p* = 0.43	1.01 (0.90, 1.13)	*p* = 0.85
General disease	0.58 (0.14, 2.32)	*p* = 0.44	0.23 (0.03, 1.68)	*p* = 0.13
Chronic medication	1.19 (0.24, 5.99)	*p* = 0.83	2.75 (0.36, 21.30)	*p* = 0.33
Diseases of musculoskeletal system in blood relatives	1.49 (0.47, 4.69)	*p* = 0.50	0.58 (0.10, 3.40)	*p* = 0.55
Past disease or trauma of musculoskeletal system	2.25 (0.52, 9.67)	*p* = 0.27	3.43 (0.26, 45.03)	*p* = 0.33
Top-level sport	6.21 (1.11, 34.80)	*p* = 0.025	0.28 (0.03, 3.07)	*p* = 0.26
	CLES = 76.17%			
Regular sporting activities	2.32 (0.54, 9.90)	*p* = 0.24		
Awareness of MSDs among dentists	0.81 (0.26, 2.53)	*p* = 0.72	1.71 (0.29, 10.00)	*p* = 0.55
Considering the range of ergonomic education as sufficient	0.88 (0.23, 3.26)	*p* = 0.84		

CLES = common language effect size.

**Table 8 ijerph-18-07662-t008:** The multivariate analysis of the influence of the followed factors on the occurrence of MSDs among the dentistry students.

	1st Year	3rd Year	5th Year			
	OR (95% CI)		OR (95% CI)		OR (95% CI)	
Gender (women)	1.37 (0.15, 12.06)	*p* = 0.78	1.13 (0.18, 7.12)	*p* = 0.90	5.50 (0.85, 35.71)	*p* = 0.061
Age	1.90 (1.01, 3.56)	*p* = 0.025	1.21 (0.75, 1.94)	*p* = 0.44	0.89 (0.56, 1.42)	*p* = 0.62
	CLES = 59.88%					
Height	0.97 (0.85, 1.10)	*p* = 0.64	1.08 (0.95, 1.23)	*p* = 0.22	1.06 (0.94, 1.19)	*p* = 0.32
Actual weight	1.03 (0.93, 1.13)	*p* = 0.61	0.93 (0.84, 1.02)	*p* = 0.10	0.99 (0.90, 1.07)	*p* = 0.74
Weight increase			0.85 (0.22, 3.27)	*p* = 0.76	2.31 (0.68, 7.87)	*p* = 0.37
Weight decrease			0.56 (0.12, 2.68)		1.72 (0.23, 12.82)	
General disease	0.89 (0.16, 5.06)	*p* = 0.90	1.02 (0.10, 10.11)	*p* = 0.99	1.21 (0.16, 9.14)	*p* = 0.86
Chronic medication	2.25 (0.47, 10.71)	*p* = 0.31	0.38 (0.05, 2.76)	*p* = 0.32	1.39 (0.15, 13.13)	*p* = 0.78
Dominant hand (right)	1.59 (0.20, 12.55)	*p* = 0.66	0.25 (0.03, 1.98)	*p* = 0.19	0.35 (0.04, 3.28)	*p* = 0.36
Diseases of musculoskeletal system in blood relatives	3.47 (0.99, 12.19)	*p* = 0.045	1.50 (0.47, 4.83)	*p* = 0.49	1.61 (0.52, 4.94)	*p* = 0.41
	CLES = 68.62%					
Past disease or trauma of musculoskeletal system	0.32 (0.03, 3.65)	*p* = 0.33	1.57 (0.38, 6.50)	*p* = 0.54	1.58 (0.33, 7.66)	*p* = 0.57
Top-level sport	2.83 (0.56, 14.23)	*p* = 0.20	1.14 (0.28, 4.76)	*p* = 0.85	5.17 (0.99, 26.94)	*p* = 0.038
					CLES = 73.91%	
Regular sporting activities	2.42 (0.42, 14.13)	*p* = 0.31	0.21 (0.04, 1.17)	*p* = 0.063	1.18 (0.28, 4.92)	*p* = 0.82
Awareness of MSDs among dentists	1.26 (0.35, 4.52)	*p* = 0.72	0.86 (0.26, 2.85)	*p* = 0.81	0.92 (0.30, 2.87)	*p* = 0.89
Considering the range of ergonomic education as sufficient			0.59 (0.16, 2.29)	*p* = 0.45	0.49 (0.12, 1.98)	*p* = 0.31

Whole model 1st year *p* = 0.20, 3rd year *p* = 0.62, 5th year *p* = 0.55; CLES = common language effect size.

**Table 9 ijerph-18-07662-t009:** The multivariate analysis of the influence of the followed factors on the newly developed MSDs among the dentistry students.

	From the 1st Year to the 5th Year	
	Newly Developed MSDs	
	OR (95% CI)	
Gender (women)	4.16 (0.41, 42.00)	*p* = 0.20
Age	0.88 (0.38, 2.07)	*p* = 0.77
Height	1.03 (0.90, 1.17)	*p* = 0.70
Actual weight	1.03 (0.94, 1.13)	*p* = 0.56
General disease	0.36 (0.02, 7.07)	*p* = 0.49
Chronic medication	0.51 (0.02, 15.96)	*p* = 0.70
Diseases of musculoskeletal system in blood relatives	1.49 (0.38, 5.88)	*p* = 0.57
Past disease or trauma of musculoskeletal system	2.46 (0.39, 15.37)	*p* = 0.33
Top-level sport	4.28 (0.58, 31.59)	*p* = 0.14
Regular sport activities	2.13 (0.35, 12.96)	*p* = 0.40
Awareness of MSDs among dentists	0.82 (0.20, 3.41)	*p* = 0.79
Considering the range of ergonomic education as sufficient	0.83 (0.17, 4.16)	*p* = 0.82

Whole model *p* = 0.68.

## Data Availability

The dataset is available upon request from the corresponding author.

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
