# Peer review of "The Development of Musculoskeletal Disorders during Undergraduate Dentistry Studies—A Long-Term Prospective Study"

_ijerph, 2021, doi:10.3390/ijerph18147662_

Round 1
Reviewer 1 Report
- Ln 66-68, The introduction oversimplifies the contributions and novelty of the study. If the only novelty of the study is conducted a follow-up study of MSDs investigation on a controlled dentistry student group in the five years, it is not sufficient.
- Ln 76, “No exclusion criteria were applied at the entry phase of the study. “ –This is not a good description. For each study, it is not possible to recruit subjects without any exclusion criteria because this is a necessary treatment to control the bias of the study’s results.
- Ln 87-88. What are the inclusion and exclusion criteria? Again, the structure of the “participant” part should be revised (from Ln 73-88). The current form is unclear and leads to many misunderstandings.
- The study design and data collection procedure are not addressed rigorously. Many factors can affect WMSDs, such as age, habit, workload, personal practices…etc. How to avoid the above potential factors influence your result?
- Ln 21-22. It will be better to state clearly about your participants (73 dentistry students and 28 general medicine students)
- It will be better to add a paragraph to address the results of Table 1.
- Why the authors only present the overall rate of the MSDs? The reader would like to know the prevalence of MSDs in different body regions.
- The Results section is not well organized, designed, and presented. (for example, why Table 3 only show the gender effect on D students? How about General medicine? Only 28 GM remained in the follow-up study is not enough)
- Discussion section. The discussion is not precise and convincing. For example, in Ln 213 to 216. Non of the measures and data is provided for comparison and discussion.
- Ln 247-251. The main finding of the present study is that the habit of doing top-level sport affected the MSDs of the selected participants. First, what is the definition of the top-level sport? Subsequently, I think that the habit of top-level sport can be a factor in the occurrence of MSDs in any occupational. If the above fact is true, based on the authors' recommendation, no job can be suggested to the people who love doing a top-level sport.
Author Response
Dear respected reviewer,
On behalf of the authors, I would like to express our gratitude for your valuable time given to our manuscript. We carefully analyzed your comments and suggestions and modified the manuscript accordingly. We really appreciate your valuable remarks, which helped us to improve our manuscript.
The replies to the particular comments are as follows:
- The novelty of this study is mainly in the long-term follow-up of the group of dentistry students because, as we state in the Introduction and Discussion, no such study has yet been conducted or has not yet been reported in the available scientific literature. Another novelty is investigating of the influence of several possible factors on the development of MSDs in the early stage of the dentistry career. We believe the gained data can contribute to a better understanding of this development. Also, our results can provide a baseline for further research in this field. The next novelty of this study is the objective measurement of students using the Spinal Mouse® device. We would like to share the acquired data in one of our following manuscripts. More about the novelty was added to the Introduction and the Discussion.
- All the students who met the inclusion criteria were invited to participate in the study. In order to avoid biasing the selection of the study participant we paid attention to the following: a clear definition of the study group (dentistry students), the choice of the good reference group (general medicine students as the control group rather than the general population). The controls should be selected independently of the exposure status. The only exclusion criterion at the beginning of the study was no previous university study in the same study program. However, it was not practically applied, because there was no such student. We did not see the necessity of more exclusion criteria. The study was planned as a long-term, prospective one. The information about the recruitment of the participants was modified in the text.
- The information about the inclusion and exclusion criteria as well as the recruitment of the students was clarified.
- In the analysis, we involved the most important factors influencing the MSDs in dental practitioners according to the current scientific literature, but only those, which are appliable to the dentistry students. We added the questionnaire as a supplementary file to show all the questions/factors. The workload is determined by the study plan and is the same for all the students. Other factors, such as age, the length of the practice, the length of the working hours, the daily number of the patients, etc., do not yet play a role in the case of the students. It is newly added to the Discussion.
- The number of general medicine students was added to the Abstract.
- Table 1 was modified to increase its clarity. We prefer to present these data in a table than in the text, which would be too long and less comprehensible. Some comments on the data presented in Table 1 were added to the Discussion.
- The table with the occurrence of pain in different body regions was added. Initially, we did not want to prolong this article too much and, thus, wanted to present these results in a separate article.
- In the Results section, we report key findings with respect to the central research questions and present secondary outcomes. The results are presented in a logical structure, starting with the number of the participants, followed by their personal characteristics, presence of the potential factors, the main results, i.e., the development of MSDs, the influence of gender, ways of solving pain and other associated factors, the analysis of the influence of the followed factors and, finally, the self-assessment of the general health. Now, we moved the general health after the presence of the factors, where it may fit better. The differences in MSDs according to gender are not presented for the general medicine students, because of a limited number of them. For the same reason, the analysis of the influence of the followed factors on the occurrence and development of MSDs was not performed in the control group. However, it does not affect the main results, as the main study group was the dentistry students.
- The Discussion was reorganized. The results of our study on dentists are referred to at several places of the Discussion. The sentence on lines 215-217 was removed, as it was out of context.
- The top-level sport, as understood in the Czech Republic, means a sporting activity that is organized in sport clubs, includes regular workouts, practice, matches, and competitions, is performed on a long-term basis, and the athletes are registered by the official, state-controlled body (e.g., the Football Association of the Czech Republic). It includes but is not limited to the professional sport. The explanation was added to the Discussion. We are not aware of the influence of the top-level sport on MSDs in other occupations; it was not the matter of this study. We can only conclude that we do not recommend the top-level athletes to study dentistry.
Reviewer 2 Report
The authors presented a long-term prospective study to investigate the occurrence of MSDs on dentistry students. This is a meaningful research, but after reading the manuscript, I find that this research has some problems that need to be clarified. Please revise the manuscript, and give explanations for these comments. These comments are listed as follows.
(1) Please provide the file of the questionnaire as the supplementary file. Otherwise, I don't know the real content of the questionnaire.
(2) For Table 1, some values are denoted to present which data of 2 years are used for comparison. For other values, no such annotations present before the p-value. Please revise this table or provide more detailed descriptions of this table.
(3) For Table 2, what are the definitions of risk and protective factors? What kinds of diseases include in the item of “Diseases of musculoskeletal system in blood relatives”?
(4) The authors also need to clarify what is “Top-level sport”. The last item (Considering the studies as psychically demanding) in Table 2 is also unclear in this table. Please explain in the article.
(5) According to study results, MSDs for general medicine students are higher than that of dentistry students. The degree of the workload of clinical work during practice should be an important factor to influence the occurrence of MSDs. If the participants answer their length of practice in the questionnaire, this quantified data can be used to realize the difference in the degree of the workload for these two groups of students. Unfortunately, no such quantified data are provided, and the conclusion of the research can only be given by inference.
(6) For Figure 1, why the GM students have a higher ratio in the development of MSDs? Do GM students have more clinical practices than dental students?
Author Response
Dear respected reviewer,
On behalf of the authors, I would like to express our gratitude for your valuable time given to our manuscript. We carefully analyzed your comments and suggestions and modified the manuscript accordingly. We really appreciate your valuable remarks, which helped us to improve our paper.
The replies to the particular comments are as follows:
(1) The translated questionnaire, along with the comments, was attached as a supplementary file.
(2) Table 1 was modified to better explain the figures.
(3) The factors were selected according to the scientific literature, being the most important factors associated with MSDs among dentists and at the same time applicable to the dentistry students. The factors, which do not yet act in students were omitted (e.g., the length of practice, the daily number of treated patients, etc.). Most of the followed factors were expected to be the risk factors according to the literature. The regular sporting activity was expected to have a protective role, i.e., to prevent MSDs. However, this was not confirmed in our sample. The “Diseases of the musculoskeletal system in blood relatives“ were not expressively defined but included any disease of the musculoskeletal system, but not trauma. According to the WHO definition, we asked about disorders affecting joints, such as osteoarthritis, rheumatoid arthritis, psoriatic arthritis, gout, ankylosing spondylitis; bones, such as osteoporosis; muscles, such as sarcopenia; the spine, such as back and neck pain or multiple body areas or systems, such as regional and widespread pain disorders and inflammatory diseases such as connective tissue diseases and vasculitis that have musculoskeletal manifestations
(4) The top-level sport, as understood in the Czech Republic, means a sporting activity that is organized in sport clubs, includes regular workouts, practice, matches, and competitions, is performed on a long-term basis, and the athletes are registered by the official, state-controlled body (e.g., the Football Association of the Czech Republic). It includes but is not limited to the professional sport. The explanation was added to the Discussion, as well as the clarification of the “Considering the studies as psychically demanding“ as the measure of subjectively perceived stress arising from the studies.
(5) We believe that there are two main reasons for the higher rate of MSDs in the general medicine students and for the increase between the first and the fifth year. (i) The general medicine students were less aware of this problem before they came to the university compared with the dentistry students. We assume that people considering dentistry to be their further occupation found the information about the MSDs occurrence among dentists in advance and were able to responsibly decide not to enter the dentistry studies, in particular, if they had already suffered from some musculoskeletal condition. (ii) As the dentistry students were aware of the MSDs occurrence among dentists, they had a chance to apply preventive matters, which, in combination with a declared sufficient range of ergonomic education, seem to be at least partly successful. Thus, we suggest a conclusion, that the initial awareness of the problem, sufficient ergonomic education, and the application of ergonomic recommendations compensated the risk factors involved during the dentistry studies, which lead to similar dynamics of the MSDs occurrence increase as in the general medicine students. An explanatory paragraph was added to the Discussion.
The question about the length of practice was not included in the questionnaire because it is known to us and is the same for all the students. At the entry point of the study, the students have no practice; in the middle of the study, the students have passed two years of the dentistry preclinical practical lessons (a total of 300 teaching hours) and one semester of introductory clinical practical lessons without patients (a total of 60 teaching hours). During the second half of the studies they spent 975 teaching hours at the clinical practical lessons. Along with these dentistry practical lessons, they have also passed through almost all the theoretical and general medicine subjects, similarly to the general medicine students, where the “dental risk factors”, such as forced unnatural position during patient treatment, do not appear. Detailed information about the workload during the studies was added to the Introduction.
(6) The explanation is included in the previous point.
Reviewer 3 Report
Estimated Authors,
Estimated Editors,
I've read with great interest the present paper from the Kapitan et al, dealing with MSD among dentistry students. The content of this paper is interesting, as well as consistent with the aims of the present journal, but it would require extensive editing before an eventual publication.
More precisely, my concerns focus on the data analysis, as this section will require the greatest overhaul.
Firstly, Authors have designed the study as a C/C one, with medical students as controls of the dentistry students. This is reasonable, as dentistry and medical students should be quite similar in terms of sociodemographic characteristics. However, during the follow up the medical students extensively dropped out from the study, and while initial ratio Case : Control was 1:0.73, at the end it was 1:0.38. In other words, the current comparisons may be quite misleading. A more appropriate approach may be redrawing the paper as a cross-section study, or - as an easier approach, by discussing very accurately such limitation in the final sections of the paper.
Second, Table 1 is confusing for two main reasons:
a) there is a lot of p values, but it is often which comparison they report on. For example: 1st row weight men: 75.9 (13.45) p < 0.01 - 78.9 (12.67) p < 0.05 - NS. With the notable exception of the last p value (i.e. NS) as the column title explains its content, the first two are not easily to associate with the original comparisons.
b) p value notation is improper. Please report the full notation of p value at least for figure > 0.001.
Third, the main shortcoming of the study:
MSD are a diverse subset of disorders, with an often very long and complex pathogenesis, with the usual interplay of several factors either individual, or external ones (including occupational ones). An appropriate way to deal with this topic requires the Authors to report the occurrence of MSD in the various body districts (i.e. upper arm, neck, back, etc.), while the present version of this paper only reports the occurrence of MSD as a whole.
Finally: you should report how univariate analysis was performed - was it a simple 2x2 table? (in this regard, please be aware that by reporting OR value, p value would be useful only as a "checkpoint" for a subsequent multivariate analysis, otherwise the analysis of 95%CI is quite appropriate. Please also be aware that the paper would be radically improved by including a multivariate analysis assuming as outcome variable the occurrence of MSD, and as explanatory variables all factors that were associated with MSD in univariate analysis.
Author Response
Dear respected reviewer,
On behalf of the authors, I would like to express our gratitude for your valuable time given to our manuscript. We carefully analyzed your comments and suggestions and modified the manuscript accordingly. We really appreciate your valuable remarks, which helped us to improve our paper.
The replies to the particular comments are as follows:
(“Firstly”) The significant decrease of the participants in the control group affected the planned analysis. Regrettably, we could not have done anything about it during the running of the study. As the control group was limited, we made the comparison of the groups only in terms of the participants characteristics and the presence of the followed factors. Other results of the general medicine students are even though presented to illustrate the overall situation. This explanation was added to two places of the Discussion.
(“Second”) a) Table 1 was modified to be better understandable.
- b) The exact p values were added to all the tables.
(“Third”) The table with the occurrence of pain in different body regions was added. Initially, we did not want to prolong this article too much and, thus, wanted to present these results in a separate article.
(“Finally”) The univariate analysis was done using the logistic regression. Initially, we did not perform the multivariate analysis, because of quite small number of the factors, which showed a statistically significant influence. Now, the multivariate analysis was calculated, and its results were added, along with the mentioning in the Methods and comments in the Discussion. The multivariate analysis was not calculated in the group of students, where the original MSDs disappeared, because of its limited size.
Round 2
Reviewer 1 Report
All of my previous comments are well addressed, and the quality of the manuscript is significantly improved. However, there is a crucial concern that needs to revise.
While the authors comment on the statistical level (p values), the authors should also calculate and interpret the magnitude/size of the effect of any statistical significance and discuss the impact of this within the context of the research and results.
Author Response
Dear respected reviewer,
On behalf of the authors, I want to thank you once again for all the effort you have put into the revision of our manuscript. Thank to your helpful comments and suggestions we were able to improve it.
The size of the effect (Cohen’s g and Common Language Effect Size) was calculated and added for the statistically significant results relevant as the defined goals of the study. Comments were added to the Methods and the Discussion.
Reviewer 2 Report
Long-term research is important to reveal the trend of developing MSDs. In this revised manuscript, the authors provided the necessary questionnaire, answered the questions well, and had more detailed explanations of their findings. I think this revised manuscript is suitable for publication.
Author Response
Dear respected reviewer,
On behalf of the authors, I want to thank you once again for all the effort you have put into the revision of our manuscript. Thank to your helpful comments and suggestions we were able to improve it.
Reviewer 3 Report
Estimated Authors,
the paper was radically improved; in my opinion it could be accepted. I'm endorsing its acceptance.
Author Response

(The authors gave the same response as above.)
